# Feasibility, Image Quality and Clinical Evaluation of Contrast-Enhanced Breast MRI Performed in a Supine Position Compared to the Standard Prone Position

**DOI:** 10.3390/cancers12092364

**Published:** 2020-08-21

**Authors:** Alfonso Fausto, Annarita Fanizzi, Luca Volterrani, Francesco Giuseppe Mazzei, Claudio Calabrese, Donato Casella, Marco Marcasciano, Raffaella Massafra, Daniele La Forgia, Maria Antonietta Mazzei

**Affiliations:** 1Department of Diagnostic Imaging, University Hospital of Siena, Azienda Ospedaliera Universitaria Senese, 53100 Siena, Italy; francesco.mazzei@ao-siena.toscana.it; 2Struttura Semplice Dipartimentale di Fisica Sanitaria, IRCCS Istituto Tumori “Giovanni Paolo II”, 70124 Bari, Italy; a.fanizzi@oncologico.bari.it (A.F.); r.massafra@oncologico.bari.it (R.M.); 3Department of Medical, Surgical and Neuro Sciences, Unit of Diagnostic Imaging, University Hospital of Siena, Azienda Ospedaliera Universitaria Senese, 53100 Siena, Italy; luca.volterrani@unisi.it (L.V.); mariaantonietta.mazzei@unisi.it (M.A.M.); 4San Rossore Breast Unit, 56122 Pisa, Italy; claudiocalabrese.it@gmail.com; 5Department of Oncologic and Reconstructive Breast Surgery, Azienda Ospedaliera Universitaria Senese, University Hospital of Siena, 53100 Siena, Italy; donato.casella@ao-siena.toscana.it; 6Unità di Oncologia Chirurgica Ricostruttiva della Mammella, “Spedali Riuniti” di Livorno, Breast Unit Integrata di Livorno Cecina, Piombino Elba, Azienda USL Toscana Nord Ovest, 57100 Livorno, Italy; marco.marcasciano@uslnordovest.toscana.it; 7Struttura Semplice Dipartimentale di Radiologia Senologica, IRCCS Istituto Tumori “Giovanni Paolo II”, 70124 Bari, Italy; d.laforgia@oncologico.bari.it

**Keywords:** breast cancer, magnetic resonance imaging, comparative studies, breast conserving therapy, local failure

## Abstract

Background: To assess the feasibility, image quality and diagnostic value of contrast-enhanced breast magnetic resonance imaging (MRI) performed in a supine compared to a prone position. Methods: One hundred and fifty-one patients who had undergone a breast MRI in both the standard prone and supine position were evaluated retrospectively. Two 1.5 T MR scanners were used with the same image resolution, sequences and contrast medium in all examinations. The image quality and the number and dimensions of lesions were assessed by two expert radiologists in an independent and randomized fashion. Two different classification systems were used. Histopathology was the standard of reference. Results: Two hundred and forty MRIs from 120 patients were compared. The analysis revealed 134 MRIs with monofocal (U), 68 with multifocal (M) and 38 with multicentric (C) lesions. There was no difference between the image quality and number of lesions in the prone and supine examinations. A significant difference in the lesion extension was observed between the prone and supine position. No significant differences emerged in the classification of the lesions detected in the prone compared to the supine position. Conclusions: It is possible to perform breast MRI in a supine position with the same image quality, resolution and diagnostic value as in a prone position. In the prone position, the lesion dimensions are overestimated with a higher wash-in peak than in the supine position.

## 1. Introduction

Many years have passed since the introduction of breast magnetic resonance imaging (MRI) in the diagnosis and monitoring of breast cancer [1,2]. Increasing evidence suggests that for many women, breast MRI provides the greatest possible accuracy, and is superior to both mammography and ultrasound. Additionally, cancers can be identified independently of the mammographic breast density. There are now clear indications for the choice of this technique in the interpretation of the images and the transfer of the results into clinical practice [3]. The guidelines prescribe the use of dedicated coils for breast MRI, with the patient in a prone position and the breasts lying within the coils. The purpose of these guidelines is to ensure an examination of good technical quality, and to provide improved diagnostic accuracy via multiparametric classification of the lesions [4] or by using the BI-RADS^®^ descriptors suggested by the American College of Radiology [5]. It has been demonstrated that breast MRIs can detect many mammary alterations that are not detected by mammograms and ultrasound, and that breast MRI has high sensitivity but variable specificity for the detection of breast cancer [4,6,7]. Even experts still diagnose a high rate of suspicious MRI lesions that have a benign diagnosis at pathology. In fact, drug intake, hormonal status, previous therapy (surgery, radiation, and chemotherapy), and risk factors (familial or genetic predisposition to breast cancer) can affect the appearance of the lesion, which can lead to lesions being classified differently. Tissue diagnosis is gaining popularity for lesions that show suspicious or indeterminate features on MRI. However, the problems related to the reassessment and histological analysis of lesions detected by MRI alone remains [7]. Targeted second-look ultrasound, which has been proposed to increase MRI specificity, makes it possible to identify correlated lesions at a pooled rate of 57.5%, considering both mass and non-mass lesions and benign and malignant lesions, with a different impact on the rates of detection [8]. This is because the standard MRI is performed in the prone position and the ultrasound in a supine position, and the displacement of the soft breast tissue causes a change in the position of some of the formations contained in it. Recently, the efficacy of MRI volume navigation has been demonstrated [9]; this method makes it possible to synchronize the MRI performed in a supine position with the ultrasound examination by coupling three pairs of markers: the multiplanar reconstructed MRI image of the corresponding ultrasound image is displayed on the monitor [10,11,12]. This new technique permits objectivity during the second-look ultrasound [10] and reduces the need for MRI-guided biopsy [11,12], but it requires the MRI to be performed in a supine position, that is the same position in which breast ultrasound is performed [13]. In addition to those already reported, supine MRI offers many other advantages: it has better correspondence with the breast lesions during conservative surgery [14,15,16,17], it clearly identifies the area of residual presence of the disease [18], it correctly identifies the volume to be subjected to radiotherapy [19,20,21], and effectively maps breast cancer in patients scheduled for oncoplastic surgery [15].

In addition to reporting the multiple advantages of supine MRI, the purpose of this study is to demonstrate that it is possible to perform the dynamic sequence of breast MRI with the patient in the supine position with the same diagnostic accuracy as that of the prone position and using the same classification criteria.

## 2. Patients and Methods

### 2.1. Study Population

After authorization by the Institutional Review Board, 302 breast MRIs were retrospectively evaluated in 151 patients who had undergone the examination in both the prone and supine position (aged 55 ± 12 years, range 24–79 years) between January 2009 and December 2015. The patients’ informed consent was waived by IRB because this is a retrospective study. In fact, all the supine MRI were performed because the patients had undergone US-guided biopsy with MRI co-registration as described in the literature [9,13]. The supine MRI were performed in the same second week of the cycle as those performed in the prone position; in subjects in menopause they were performed no more than ten days after those in the prone position. Subjects who had undergone the prone MRI in a different center, and did not have the entire examination available in a digital format, were excluded from the analysis. The numerical representativeness of the sample was verified on the basis of previous studies [8]. The prevalence of locating additional lesions in breast MRIs performed in the prone position is 25.3% of total MRIs. Therefore, establishing a level of significance of 5% and an estimate precision of 8%, the minimum sample size is 113 subjects [22]. Consequently, the sample of 120 subjects with MRIs in both the prone and supine position that were available to us was proven to be representative of the general population.

### 2.2. MRI in the Prone Position

The MRI examinations were carried out using two 1.5 T MR scanners from Achieva Philips Healthcare and Signa HDtx, General Electric Healthcare. The following parameters were used: (1) THRIVE SPAIR sequence (T1-weighted high resolution isotropic volume examination, spectral attenuated inversion recovery), 130 axial 1-mm partitions (TR/TE = 4.7/2.3 ms; FA = 10° FOV = 320 mm; matrix = 300 × 300 mm; time 9′40″) with 116-s time resolution; 1 pre- and 4 postcontrast (Achieva Philips Healthcare); (2) VIBRANT sequence (volume imaging for breast assessment), 130 axial 0.8 mm partitions (TR/TE = 5.8/2.8 ms; FA = 10° FOV = 320 mm; matrix = 300 × 300 mm; time 8′33″) with 106-s time resolution; 1 pre- and 4 postcontrast phases (Signa HDxt, General Electric Healthcare). Dedicated 7 and 8-channel coils supplied by the manufacturer were used in the scanners, respectively.

### 2.3. MRI in the Supine Position

Using the same MR scanners described above, the dynamic sequences of the breast in the free-breathing subject were acquired with the subject in the supine position: (1) THRIVE SPAIR sequence, 180–220 axial 1 mm partitions (TR/TE = 4.7/2.3 ms; FA = 10° FOV = 420 mm; matrix = 330 × 420 mm; time 6′30″) with 120” time resolution, 1 pre- and 3 postcontrast (Achieva Philips Healthcare); (2) LAVA sequence (liver acquisition with volume acceleration), 160–200 axial 0.8-mm partitions (TR/TE = 4.5/2.1 ms; FA = 10° FOV = 460–360 mm; matrix = 320 × 320 mm; time 10′15″) with 123″ time resolution; 1 pre- and 4 postcontrast phases (Signa HDxt, General Electric Healthcare).

Body coils of the same type used for study of the abdomen were used in both scanners. The subject lay on the bed in a supine position with arms stretched above the head, placing pads between the body and the coil to prevent any compression of the breast tissue caused by the weight of the coil, which was attached to the body using straps as shown in Figure 1.

### 2.4. Contrast Medium

In both the MRI procedures, after the first pre-contrast sequence, the same intravenous contrast medium was administered at a dose of 0.2 mL/kgbw of Gd-BOPTA (Bracco Imaging), followed by approximately 20 mL of saline solution.

### 2.5. Recommendations before the MR Examination

During both examinations, but particularly during the MRI in the supine position, subjects were asked to avoid taking deep and frequent breaths and to maintain a regular breathing rhythm.

### 2.6. Image Evaluation

All the MRI examinations performed in the supine position were compared with those performed in the standard prone position. The presence of artifacts that could restrict the diagnostic value of the MR examination was assessed by consensus by two radiologists experienced in breast imaging. The signal-to-noise Ratio (SNR) and contrast-to-noise Ratio (CNR) were calculated for all pairs of examinations considering six regions of interest. Glandular tissue and fatty tissue were used to calculate the CNR in both examinations. Moreover, the images were compared by considering the number of lesions detected in each examination, and they were divided into monofocal, multifocal and multicentric lesions, as described in the literature [5]. The absence of raw data from the MRI performed in the prone position was an exclusion criterion. The maximum extension of the index lesion was then compared, using multiplanar reconstructions when necessary.

The comparison of the characteristics of the index lesion considered the form of the lesion, and the edges and homogeneity of the contrast medium. Using a specific software for each console (MR Extended WorkSpace, Philips Healthcare and Advantage Workstation VolumeShare 2, General Electric Healthcare) the percentage-time intensity curves for specific regions of interest (ROI) in the area of greatest impregnation of contrast medium in the index lesion were constructed to compare the type of curve obtained in the two different positions. Through the colorimetric maps of the maximum contrast impregnation, an analysis of the maximum peak of distribution was made to decide where to position the ROI and to highlight all the areas of maximum impregnation. This also made it possible to compare the variation in the vascularization of the index lesion in the two different positions.

After collecting all the necessary parameters, two radiologists who were very experienced in breast imaging (Op_A and Op_B) used the classification proposed by Baum et al. [4] and the BI-RADS^®^ criteria [5] for the classification of the breast lesions, using random numbers generated by computer and independently analyzing the MRI examinations.

### 2.7. Statistical Analysis

The detected parameters and measures were summarized in terms of median and interquartile range (i.e., the interval between the first and third quartiles of the empirical distribution). The parameters and measures obtained in the two different positions were compared using the Wilcoxon rank sum test, which tests the significance of the difference of the means in paired observations [23]. A result was considered statistically significant when the *p*-value was less than 0.05. MATLAB R2018a (MathWorks, Inc., Natick, MA, USA) software was used for the statistical analysis.

## 3. Results

After the exclusion of 62 breast MRIs in 31 patients, due to the absence of raw data for the MRI examinations in the prone position, 240 breast MRIs were compared in 120 subjects (aged 55 ± 12 years, range 24–79 years). None of the investigations performed in the supine position were excluded from the analysis due to the presence of artifacts. In 11% (13/120) of the MRIs in the supine position, the dynamic sequence demonstrated incomplete fat suppression evident in the native sequences. This alteration had no effect on the subtracted sequences.

The SNR measured in the prone and supine position was 87.91 (65.59–131.79) and 91.91 (69.35–132.12), respectively, whereas the CNR was 41.35 (29.26–52.77) and 34.55 (27.44–50.78). Both the SNR and CNR measured in the prone and supine positions showed no significant differences (*p* = 0.47 and *p* = 0.13).

The analysis indicated the presence of 134 MRIs with monofocal lesions (U), 68 with multifocal lesions (M) and 38 with multicentric lesions (C). Ten MRIs showed bilateral breast lesions: 3 U, 1 M and 1 C. Histopathology revealed 66 malignant lesions and 54 benign lesions, as listed in Table 1 and Table 2.

There was no difference between the number of lesions found in the prone and supine examinations. No supine examination was excluded due to the presence of respiratory artifacts, just as in the prone examination.

The measurements of the index lesion made by Op_A and Op_B in the prone position and in the supine position are shown in Figure 2 with the lesion dimensions detected by histopathology.

There was no significant difference detected between the intraobserver (Table 3a) and interobserver (Table 3b) measurements. However, the statistical analysis of the dimensions of the different positions revealed a significant difference in the extension of the lesions between the prone and supine position (Table 3c). There was also a significant difference in the distribution of the measurements detected by the histological examination as compared to those detected in the prone position, but not with those detected in supine position (Table 3d).

Table 4 illustrates the statistical analysis used to compare mass-like and non-mass-like lesions, and shows the different MRI measurements in the prone and supine position compared to the histopathology.

Table 5 presents the statistical analysis resulting from grouping different histopathologic types of lesion and the effect on lesion measurement during MRI in the prone and supine position. Examples are shown in Figure 3 and Figure 4.

The interpretations of the two radiologists showed no significant differences in the results of the BI-RADS^®^ classification [5] of the lesions detected in the prone and supine positions (*p* = 0.792). Even when the parameters suggested by Baum et al. [4] were used for classification, no significant differences were observed (*p* = 0.942). Although the absolute values of the curves for specific ROI cannot be compared in the two positions due to distortion of the mammary parenchyma, no significant difference was observed in the morphology of the curves pursuant to the classification by type: I, II, III [4,5]. Nevertheless, a significant reduction in the values of the initial enhancement and wash-in of the lesions studied in the supine position as compared to prone was recorded (*p* = 0.046). This difference is more marked in the distribution of the vascularization peak using the colorimetric maps, as shown in Figure 5, Figure 6 and Figure 7.

More specifically, Figure 7 shows the greater signal intensity (in red) of the breast lesion, which represents a higher contrast impregnation peak compared with the same lesion in the image acquired in the supine position.

## 4. Discussion

The results of our study demonstrate that it is possible to perform breast MRI in the supine position with the same diagnostic value as in the prone position. The image quality, spatial resolution and the absence of artifacts makes the examination diagnostic, as confirmed by comparison with the examination performed in the standard prone position (see Appendix A).

To the best of our knowledge, this is the first published study that uses the same patient to compare MRI images taken in the prone and supine position, which is the only way of investigating the presence of potential variations. Our study employed a high-resolution matrix with nearly isotropic voxel of approximately 1 mm, and used a T1-weighted dynamic sequence with fat suppression at high resolution in both positions. The performance of the dynamic technique in the supine position enabled us to make electronic subtractions, as for the prone position, and to assess the vascularization of the breast lesions over time. The comparison of the dynamic behavior in the two positions makes this study unique. This opens up the prospect of acquiring further knowledge regarding the vascularization of breast lesions and the real extension of the tumors. Indeed, one of the parameters that varies most between the prone to a supine position is the contrast impregnation peak, or wash-in. The different position appears to alter the impregnation peak, and thus the ROI is positioned in the area of maximum contrast impregnation in both patient positions. This alteration is shown more clearly in the colorimetric maps of the contrast impregnation peaks in the two different positions, since these consider the entire volume of the lesion rather than the individual region of interest. They demonstrate that the maximum peak of vascularization is reduced in all lesions in the transition from the prone to a supine position. This led us to the conclusion that, not only was the precise analysis of the ROI an indicator of a reduced maximum peak, but the analysis of the overall reduction using the colorimetric maps provided definitive confirmation of the different vascularization in the prone and supine positions.

Although these differences can be minimized through the use of computer-aided detection systems in MRI, as documented in the literature for other breast imaging methods [24,25,26,27,28,29], our study demonstrated that there is a significant difference in all the breast lesions in the images from the prone and the supine position with the exception of the fibroadenomas, the single papilloma, the lymph nodes and a silicoma, benign formations of the breast and invasive ductal carcinoma. In the remaining cases, a significant variation in the extension was observed, which generally consisted of a reduction in the extension in the prone compared to the supine position.

In the light of this data, our research opens up a new possibility for the evaluation of mammary neoplasia, since it improves the correlation between the position of the breast lesion and the position that the surgeon has to consider on the operating table. This means that the surgeon no longer has to hypothesize the displacement effect on the breast lesion as is the case for the prone position examination, but they can actually observe the real position of the breast lesion or lesions in relation to the nipple and the pectoral muscle [30].

In practice this means that the comparison of the MRI examinations of the breast in the prone and supine positions could alter the planning of surgical operations. A reduction in the involvement of the margins of the breast lesion can plausibly be expected due to the absence of displacement in the examination performed in the supine position and the position of the patient during the operation, since a reduced extension of the lesion in the supine position has been demonstrated.

We also demonstrated that the same classification criteria used for the examination performed in the prone position can be used, namely the BIRADS^®^ criteria [5] and those suggested by Baum et al. [4], as they have the same diagnostic value.

Although the examination was performed on the same patient in both the prone and supine positions, we did not include an estimate of the spatial displacement of the lesions in our objectives. Indeed, although it might seem easy to estimate the displacement of the breast lesions by changing from the prone to the supine position, on the contrary, the models proposed and published to date [31,32,33] have not considered the same examination technique in the transition to the supine position, or have made only one assessment of the displacement. We believe that the results would have been statistically more consistent if the measurements had been repeated several times so as to assess the error in the estimate of the lesion position. Consequently, we did not take this aspect into consideration since it was not possible in our retrospective study.

Another aspect that was not included in the objectives of this study is the difference in the practices of the radiologists who perform the breast MRI, for example, although this method can be applied to all patients, it is particularly difficult in the presence of breast hypertrophy when the breast tissue extends beyond the front edges of the thorax.

Therefore, the application of this different position in preoperative staging, appears to be restricted to the performance of objective second-look ultrasounds with MRI co-registration, and to the performance of US-guided biopsies with MRI co-registration. The examination could also be restricted to women whose physical constitution precludes the use of the dedicated coil for the study of the breast because the dimensions of the gantry are insufficient.

## 5. Conclusions

Our study demonstrates that it is possible to perform a breast MRI in the supine position and that it has the same diagnostic value as an MRI in the prone position. Moreover, by changing from using the MRI examination in the prone to a supine position the extension of the breast lesions appears to be significantly reduced, with the exception of certain benign lesions.

Finally, given the correlation between the position of the breast lesion in the supine MRI and the position that the surgeon has to consider on the operating table, when the preoperative examination is performed in the supine position this has the advantage of reducing the involvement of the margins during surgery.

## Figures and Tables

**Figure 1 cancers-12-02364-f001:**
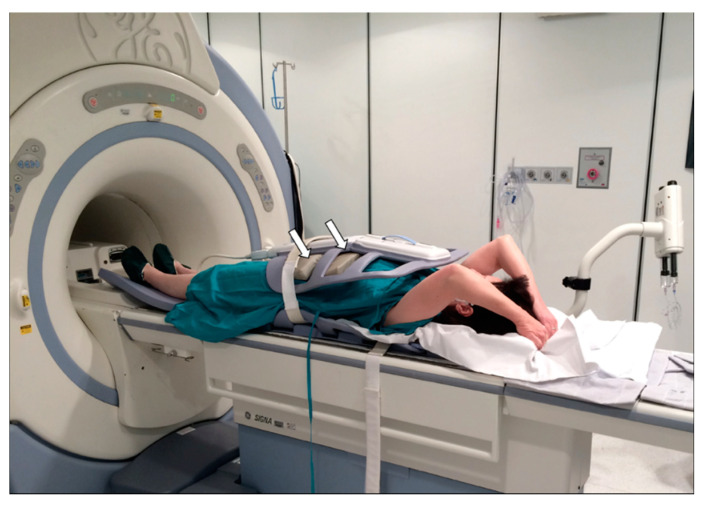
Patient in supine position for dynamic magnetic resonance imaging (MRI) of the breast using a body coil, which is the same as that used for the examination of the abdomen. Note the presence of a pad between the patient’s abdomen and the coil at the level of the strap (arrows). The portion of the coil adjacent to the breast is not hooked to the strap to prevent any possible distortion of the mammary parenchyma during the examination.

**Figure 2 cancers-12-02364-f002:**
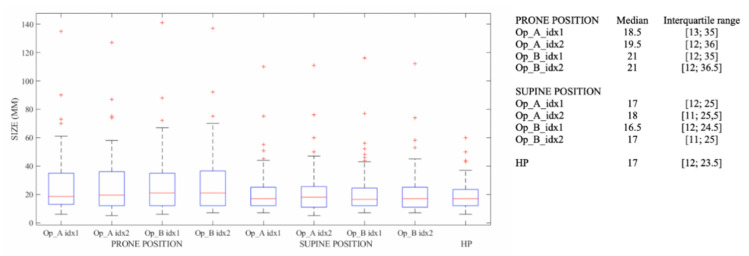
Measurements of the index of the lesion dimension with MRI in the prone position and in the supine position and the lesion dimensions detected by histopathology.

**Figure 3 cancers-12-02364-f003:**
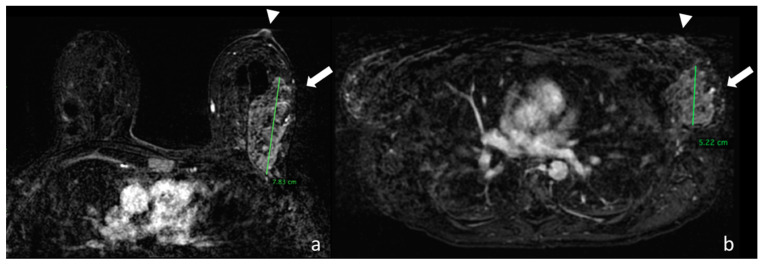
(**a**) Image taken from the first dynamic sequence performed in the prone position showing the nipple area (arrowhead). At the confluence of the external quadrants of the left breast there is a multicentric pathological impregnation of the contrast medium, which measures around 7.83 cm, which histological analysis identified as a lobular carcinoma infiltrating and in situ (arrow). (**b**) Image taken from the first dynamic sequence performed in the supine position showing the nipple area (arrowhead) in the same patient. At the confluence of the external quadrants of the left breast there is the same multicentric pathological impregnation of the contrast medium, but with a smaller extension of around 5.22 cm.

**Figure 4 cancers-12-02364-f004:**
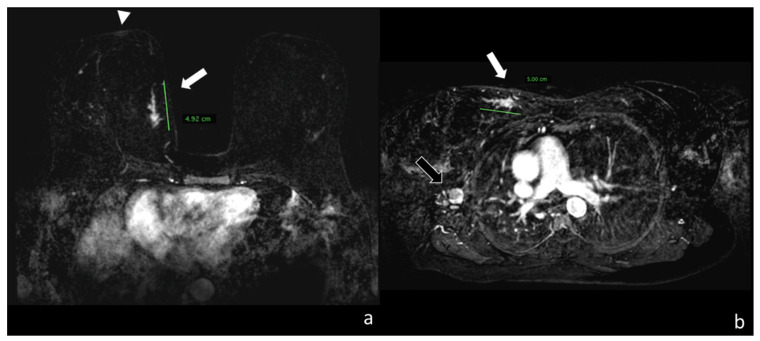
(**a**) Image taken from the first dynamic sequence performed in the prone position showing the nipple area (arrowhead). At the confluence of the internal quadrants of the right breast it shows a linear pathological impregnation of the contrast medium measuring around 4.92 cm, which histological analysis identified as a ductal carcinoma infiltrating and in situ (white arrow). (**b**) Image taken from the first dynamic sequence performed on the same patient in the supine position, showing how the lesion is localized in the upper internal quadrant of the right breast (see the mediastinum) with a similar extension of around 5.00 cm (white arrow). Note how in the supine position, the secondary localization in the same layer appears in the site of the right axilla (black arrow).

**Figure 5 cancers-12-02364-f005:**
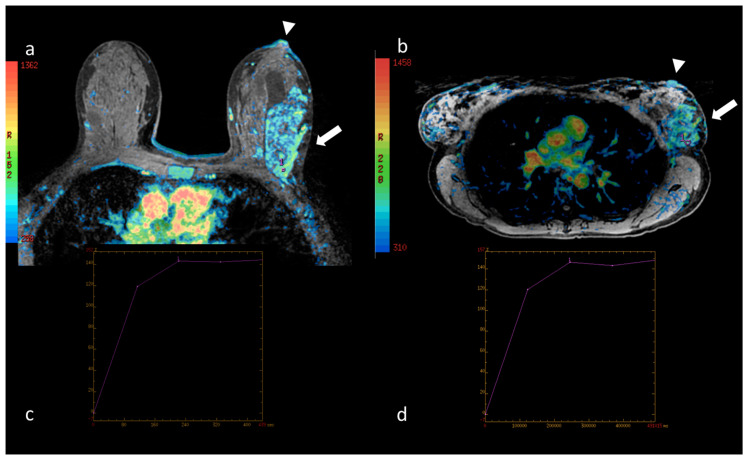
(**a**) Native image of the first dynamic sequence performed in the prone position superimposed with the colorimetric map showing maximum contrast impregnation (see the reference side bar) in the nipple area (arrowhead). At the confluence of the external quadrants of the left breast, the site of the neoplasia described in Figure 3, the distribution of the contrast medium is shown (arrow); in the area of maximum impregnation a region of interest (ROI) has been drawn marked by the number 1. The corresponding percentage/time intensity curve (au/sec, arbitrary units/second) is shown in (**c**). (**b**) Native image of the first dynamic sequence performed in the supine position superimposed with the colorimetric map. In this position, the dynamic curve of the breast lesion was evaluated in a ROI marked by the number 1. The corresponding percentage/time intensity curve (au/sec) is shown in (**d**). In both positions, the curve is a type I with the same maximum peak values in both the early and late phases, although the ROIs are of different sizes and positioned in different areas of the lesion.

**Figure 6 cancers-12-02364-f006:**
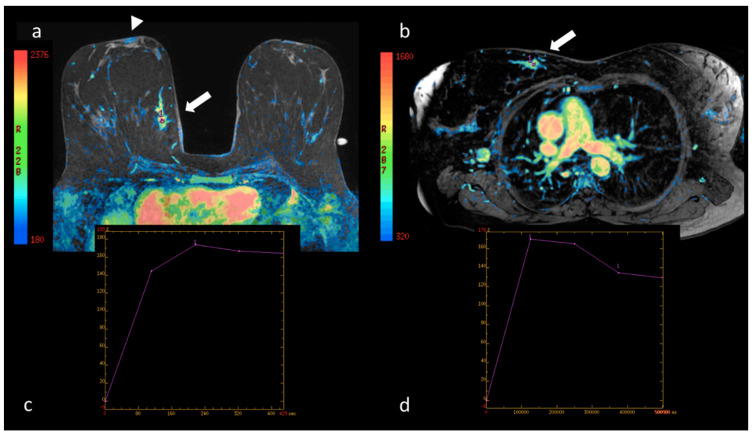
(**a**) Native image of the first dynamic sequence performed in the prone position superimposed with the colorimetric map showing maximum contrast impregnation (see the reference side bar) in the nipple area (arrowhead). At the confluence of the internal quadrants of the right breast, the site of the neoplasia described in Figure 4, the distribution of the contrast medium is shown (arrow); in the area of maximum impregnation a region of interest (ROI) has been drawn marked by the number 1. The corresponding percentage/time intensity curve (au/sec) is shown in (**c**). (**b**) Native image of the first dynamic sequence performed in the supine position superimposed with the colorimetric map. In this position, the dynamic curve of the breast lesion was evaluated in a ROI marked by the number 1. The corresponding percentage/time intensity curve (au/sec) is shown in (**d**). In both positions the curve is a type III with the same maximum peak values in the early phase; in (**d**) the late-phase wash-out appears more accentuated.

**Figure 7 cancers-12-02364-f007:**
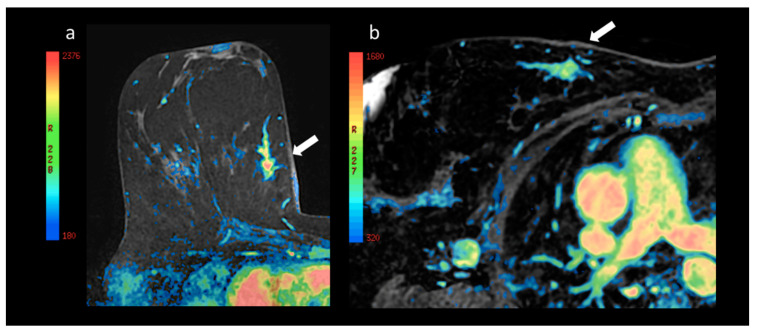
(**a**) Enlargement of the lesion at the confluence of the internal quadrants of the right breast with superimposition of the colorimetric map already shown above in Figure 6, and (**b**) the lesion corresponding to the ipsilateral upper internal quadrant in the prone and supine position (arrow), respectively. Note the different distribution of the colorimetric map with a prevalence of elevated maximum increase values after the administration of contrast in the prone position (**a**) as compared with the supine position (**b**).

**Table 1 cancers-12-02364-t001:** Benign lesions detected by histopathology.

Type	N
FA	19
FBS	9
LYMP	6
FLO CRO	4
NA	4
PAP	3
SCL AD	3
LCIS	2
ADH	2
SIL	1
PAP M	1
	54

Abbreviations: FA, fibroadenoma; FBS, fibrosclerosis; LYMPH, intramammary lymph node; FLO CRO, chronic inflammation; NA, fat necrosis; PAP, single papilloma; SCL AD, sclerosis adenosis; LCIS, lobular carcinoma in situ; ADH; atypical ductal hyperplasia; SIL, silicoma; PAP M, multiple papilloma.

**Table 2 cancers-12-02364-t002:** Malignant lesions detected by histopathology.

Type	N
IDC	27
IDC + DCIS	19
DCIS	10
ILC	5
IDC + ILC	2
ILC + LCIS	2
DCIS + LCIS	1
	66

Abbreviations: IDC, invasive ductal carcinoma; DCIS, ductal carcinoma in situ; ILC, invasive lobular carcinoma; LCIS, lobular carcinoma in situ.

**Table 3 cancers-12-02364-t003:** *p*-values of lesion dimensions. (**a**) Intraobserver Test; (**b**) Interobserver Test; (**c**) Test P vs. S; (**d**) Test idx1–2 vs. HP.

**(a) Intraobserver Test**									
	**P**				**S**				
	**idx1**		**Idx2**	**Sign**	**idx1**		**Idx2**		**Sign**
Op_A	18.5 [13.0; 35.0]		19.5 [12.0; 36.0]	-	17.0 [12.0; 25.0]		18.0 [11.0; 25.5]		-
OP_B	21.0 [12.0; 35.0]		21.0 [12.0; 36.5]	-	16.5 [12.0; 24.5]		17.0 [11.0; 25.0]		-
**(b) Interobserver Test**									
**Position**		**Op_A (idx1 + idx2)**			**Op_B (idx1 + idx2)**			**Sign**	
P		19.0 [12.0; 35.0]			21.0 [12.0; 35.5]			-	
S		17.0 [11.0; 25.0]			17.0 [12.0; 25.0]			-	
**(c) Test P vs. S**									
		**P**			**S**			**Sign**	
Op_A (idx1 + idx2)		19.0 [12.0; 35.0]			17.0 [11.0; 25.0]			*	
Op_B (idx1 + idx2)		21.0 [12.0; 35.5]			17.0 [12.0; 25.0]			*	
		20.0 [12.0; 35.0]			17.0 [12.0; 25.0]				
**(d) Test idx1–2 vs. HP**									
**Position**	**Op_A**				**Op_B**				**HP**
	**idx1**	**Sign**	**idx2**	**Sign**	**idx1**	**Sign**	**idx2**	**Sign**	
P	18.5 [13.0; 35.0]	*	19.5 [12.0; 36.0]	*	21.0 [12.0; 35.0]	*	21.0 [12.0; 36.5]	*	17.0 [12.0; 23.5]
S	17.0 [12.0; 25.0]	-	18.0 [11.0; 25.5]	-	16.5 [12.0; 24.5]	-	17.0 [11.0; 25.0]	-	17.0 [12.0; 23.5]

Abbreviations: S, supine position; P, prone position; Op_A, operator A; Op_B, operator B; idx1, first measurements of index lesion; idx2, second measurements of index lesion; HP, measurement detected by histopathology; * *p*-value Wilcoxon rank sum test < 0.05.

**Table 4 cancers-12-02364-t004:** Median and interquartile range of lesion dimensions considering mass-like and non-mass like lesions.

Type	Sample Size	P	S	HP	P vs. HP	S vs. HP
Mass-like	62	13.0 [10.0; 21.0]	13.0 [10.0; 19.0]	14.0 [11.0; 21.0]	-	-
Non-mass-like	58	28.5 [21.0; 42.0]	23.0 [16.0; 34.0]	20.0 [13.0; 25.0]	*	*

Abbreviations: S, supine position; P, prone position; Op_A, operator A; Op_B, operator B; idx1, first measurements of index lesion; idx2, second measurements of index lesion; HP, measurement detected by histopathology; * *p*-value Wilcoxon rank sum test < 0.05.

**Table 5 cancers-12-02364-t005:** Median and interquartile range of lesion dimensions grouped in different histopathologic types.

Type	Sample Size	P	S	HP	P vs. HP	S vs. HP
1	25	24.0 [16.5; 28.0]	20.0 [13.5; 24.5]	19.0 [12.8; 21.0]	*	-
2	27	11.0 [9.0; 16.0]	11.0 [9.0; 15.5]	12.0 [8.3; 18.8]	-	-
3	27	15.0 [12.0; 20.0]	15.0 [11.0; 21.0]	13.0 [12.0; 22.8]	-	-
4	22	34.5 [20.5; 43.0]	23.0 [15.0; 36.0]	24.5 [17.0; 32.0]	*	-
4 + 5	41	37.0 [22.0; 47.5]	25.0 [16.0; 36.5]	23.0 [13.0; 28.3]	*	*
5	19	42.0 [23.0; 52.5]	33.0 [16.5; 37.0]	17.0 [9.0; 25.0]	*	*

Abbreviations: Type 1, ADH, FBS, FLO CRO, LCSI, NA, PAP M, SCL AD; Type 2, FA, LYMP, PAP, SIL; Type 3, IDC; Type 4, DCIS, DCIS + LCIS, ILC, IDC + ILC, ILC + LCIS; Type 5, IDC + DCIS; * *p*-value Wilcoxon rank sum test < 0.05.

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
