# Peer review of "Feasibility, Image Quality and Clinical Evaluation of Contrast-Enhanced Breast MRI Performed in a Supine Position Compared to the Standard Prone Position"

_cancers, 2020, doi:10.3390/cancers12092364_

Round 1
Reviewer 1 Report
The authors describe a single-institution comparison of prone and supine breast MRI images. Studies such as this can provide important data for clinical practice and for the design of randomised clinical trials. However, due to the major criticism (below), this manuscript can only be considered for publication after significant revision.
The major criticism of the manuscript is the lack of clarity regarding the sample size. If the sample size was calculated based on a pre-specified hypothesis and power calculation, this should be stated and details given. However, the impression is that this is the result of convenience sampling, that is selection was made on the basis of availability of data to the investigators. There are two problems with convenience samples: (i) the patients might not be representative of the wider population, and (ii) hypothesis tests should be avoided. If this is a sample of convenience then this should be clearly stated, and the results of the hypothesis tests (p values) removed. Furthermore, the conclusions should be restrained.
Minor criticisms (not a complete list):
Line 40: "To assess feasibility..." Is this important? If so, how was this assessed?
Line 45: "in a double-blind" Need details - not mentioned elsewhere.
Line 83: "interval" Is this the 95% CI? Or range?
Line 147: "randomly and independently analyzing the MRI examinations." Need details of the randomisation procedure.
Line 191: This Table is confusing.
Line 201: This Table (without p values) is useful, but the number of decimal places seems excessive. The number of patients in each group should be shown. Also, dimensions should be stated.
Line 347: This Graph is useful and can be used instead of all the p values.
Author Response
The authors describe a single-institution comparison of prone and supine breast MRI images. Studies such as this can provide important data for clinical practice and for the design of randomised clinical trials. However, due to the major criticism (below), this manuscript can only be considered for publication after significant revision.
Point 1.The major criticism of the manuscript is the lack of clarity regarding the sample size. If the sample size was calculated based on a pre-specified hypothesis and power calculation, this should be stated and details given. However, the impression is that this is the result of convenience sampling, that is selection was made on the basis of availability of data to the investigators. There are two problems with convenience samples: (i) the patients might not be representative of the wider population, and (ii) hypothesis tests should be avoided. If this is a sample of convenience then this should be clearly stated, and the results of the hypothesis tests (p values) removed. Furthermore, the conclusions should be restrained.
Response 1. Wethank the reviewer for having given us the opportunity to clarify the way in which we calculated the sample size. Ours is a sample of convenience, to the extent that it is made up of all the MRI results available in our database from 2009 to 2015, obtained in a consecutive manner and without any selection criterion correlated to the phenomenon being studied.
Nevertheless, we should emphasize the fact that we considered the problem of whether the sample was representative of the wider population. Since no studies have been published on the variability of the differences in dimension between lesions measured in prone and supine MRI breast exams, we were unable to estimate the size of the sample in comparison to this parameter, which was the subject of analysis.
It is, however, important to point out that in one of our earlier studies, cited at no. 8 of the bibliography, we observed that the expected prevalence of an additional lesion in breast MRI in prone position appears in 25.3% (490/1930) of total MRIs. Therefore, establishing a level of significance of 5% and an estimate precision of 8%, the minimum sample size is 113 subjects [see Kasiulevičius, V.; Šapoka, V.; Filipavičiūtė, R. Sample size calculation in epidemiological studies. Gerontologija. 2006, 7 (4), 225-231.]. Consequently, the sample of 120 subjects in both prone and supine position available to us proves to be representative of the general population.
We did not think it opportune to include this calculation in the text to avoid complicating the reading of the article. Nevertheless, we have accepted the reviewer’s just observation by specifying that this verification was indeed made, and have added this phrase in the text: “The numerical representativeness of the sample was verified on the basis of previous studies.”
Point 2.Minor criticisms (not a complete list):
Line 40: "To assess feasibility..." Is this important? If so, how was this assessed?
Response 2. The guidelines published in literature[3] prescribe the performance of breast MRI using dedicated coils with the patient in prone position. No manufacturer of MRI equipment indicates the possibility of performing breast MRI with coils other than the dedicated ones.
For us it is important to state that it is possible to perform breast MRI in supine position since this has been our experience, and the comparison with the MRI performed in prone position in the same patient makes the similarities and differences of the two postures even more evident.
We have been able to demonstrate that respiratory artifacts do not prevent the performance of MRI in supine position and that the same spatial and temporal resolution can be obtained using coils other than breast coils.
Point 3. Line 45: "in a double-blind" Need details - not mentioned elsewhere.
Response 3. We have replaced this word in line 45 with “independent”. The description is contained in the paragraph “Image Evaluation”.
Point 4. Line 83: "interval" Is this the 95% CI? Or range?
Response 4. We thank the reviewer for the suggestion and we changed “interval” to “range” in the text.
Point 5. Line 147: "randomly and independently analyzing the MRI examinations." Need details of the randomisation procedure.
Response 5. We used a sample of patients extracted randomly using random numbers generated by computer.
Point 6. Line 191: This Table is confusing.
Response 6. We have altered Table 3 to detail the median and interquartile range of the comparisons made.
Point 7. Line 201: This Table (without p values) is useful, but the number of decimal places seems excessive. The number of patients in each group should be shown. Also, dimensions should be stated.
Response 7. As suggested by the reviewer we have reduced the numbers to a significant decimal figure and indicated the number of patients in each group.
Point 8. Line 347: This Graph is useful and can be used instead of all the p values.
Response 8. As indicated in point 6, we have detailed the median and interquartile range referring to the graph and removed the p-values of Tables 3, 4 and 5, leaving only the indication of the significance of the result.
Reviewer 2 Report
The authors present a comparison between MRIs obtained in the supine position compared to the prone position. While interesting, the paper is difficult to read and particularly lacking in analysis on the results.
The introduction is very short and does not appear to introduce the topic fully. What is the state-of-the-art in terms of breast MRI positioning? What techniques are limited in orientation? Is there any reason to expect that prone and supine MRI should be different? Is either preferred by patients/clinicians?
For the population, having both MRIs performed was an inclusion criterion. How would this decision have been made (to undergo the US-guided biopsy with MRI co-registration) and how does this bias the study population?
For example, "There was no difference in the number of lesions between the prone and supine examinations": were all lesions identified in prone identified in supine and vice versa? Is this expected? Is there any reason to assume this wouldn't be the case?
The tables are very difficult to read and extremely poorly described.
Very little analysis is presented, mostly only p-values or population-wide averages. Pairwise comparisons would be much more interesting exploiting the fact that the same patient is scanned in both orientations. It is very difficult to determine any differences between the two orientations when only p-values and global IQRs are presented.
Author Response
The authors present a comparison between MRIs obtained in the supine position compared to the prone position. While interesting, the paper is difficult to read and particularly lacking in analysis on the results.
Ponte 1. The introduction is very short and does not appear to introduce the topic fully. What is the state-of-the-art in terms of breast MRI positioning? What techniques are limited in orientation? Is there any reason to expect that prone and supine MRI should be different? Is either preferred by patients/clinicians?
Response 1. We agree with the reviewer and we have duly added two phrases in the introduction to explain the concept that the standard breast MRI is performed with the patient in prone position as indicated in the guidelines.
Point 2.For the population, having both MRIs performed was an inclusion criterion. How would this decision have been made (to undergo the US-guided biopsy with MRI co-registration) and how does this bias the study population?
For example, "There was no difference in the number of lesions between the prone and supine examinations": were all lesions identified in prone identified in supine and vice versa? Is this expected? Is there any reason to assume this wouldn't be the case?
Response 2.Certainly: one of the inclusion criteria was precisely that the same patient had undergone both prone and supine MRIs so as to make the comparison. There would have been no other indication for performing an MRI in supine position except for surgical purposes, that is to perform the US-guided biopsy with MRI co-registration.
Although this might seem a bias of the study population, it was the only acceptable criterion for performing a supplementary MRI exam in supine position.
The result that, in the MRI exam performed in supine position the same lesions as in prone position were identified, was anything but predictable. In fact the guidelines lay down that the exam should be performed in prone position since it is considered that in other positions the exam might not be diagnostic. Moreover, all the manufacturers of MRI machines supply dedicated coils for study of the breast designed for the patient lying in prone position.
Point 3.The tables are very difficult to read and extremely poorly described.
Response 3.We have altered Table 3 to detail the median and interquartile range of the comparisons made. We have also provided a more detailed description of the results shown in Table 3.
Point 4.Very little analysis is presented, mostly only p-values or population-wide averages. Pairwise comparisons would be much more interesting exploiting the fact that the same patient is scanned in both orientations. It is very difficult to determine any differences between the two orientations when only p-values and global IQRs are presented.
Response 4.As indicated in the paragraph “Statistical analysis”, the non-parametric test used is the Wilcoxon rank sum test, which is commonly used as an alternative to the parametric test for paired data. The statistics of the Wilcoxon rank sum test are in fact calculated on the difference between coupled pairs of observations. We have clarified this in the “Statistical analysis” paragraph and in the captions we have provided the full name of the test used. Nevertheless, to avoid weighing down the presentation of the results we decided not to display the distribution of the differences but only the distribution of the measurements detected in the two positions, which do however demonstrate that the measurements detected in supine position are on average smaller than those detected in prone position.
Reviewer 3 Report
Dear authors,
The presented manuscript is very well written and mostly well presented. I do find that however some further analysis should be completed. Please find my comments below.
Line 133: what is footnote 5?
One extension that should be applied to the data concerns volume (or at least area calculation) of the tumour. Volumetric data should have been considered to compare the tumour volume as calculated in exams in supine versus prone positions – instead of just comparing the largest axis of the tumour in a single image for the two image types.
p-value test is often used with lower-case “p”
figures 3 and 5 – impossible to analyse graphs in c) and d) – numbers cannot be seen and lines are very faint, also axes information seems to be missing – please improve these graphics.
Following the suggestions above, please provide a systematic study analysing differences in size of ROI between supine and prone exams.
Line 254: what do you mean by “ROI marked by the number 1”? There is no number 1 marked in the figure
Figure 2 is followed by figure 4, and then figure 3 is followed by figure 5 - why do you not use a regular sequence of numbers?
Lines 299-305: a number of observations seem to have been made, but there is no systematic presentation of these in the results. Overall it feels like the data can be a lot more scrutinised and more information should be extracted from the data available to the study.
Author Response
Dear authors,
The presented manuscript is very well written and mostly well presented. I do find that however some further analysis should be completed. Please find my comments below.
Point 1.Line 133: what is footnote 5?
Response 1. The mistake was corrected, inserting the bibliographical reference [5]
Point 2. One extension that should be applied to the data concerns volume (or at least area calculation) of the tumour. Volumetric data should have been considered to compare the tumour volume as calculated in exams in supine versus prone positions – instead of just comparing the largest axis of the tumour in a single image for the two image types.
Response 2.We are in complete agreement with the reviewer that it would theoretically be best to provide the tumour volume data. However, in practice this is not applicable for two main reasons. Firstly, many lesions are irregular in form, especially non-mass and multicentric lesions. Secondly, the calculation of volume using manual contouring would take a very long time and for this reason is not performed in clinical practice. Obviously the volume does not change with the change in position; what we wish to stress is that the maximum extension of the lesion changes depending on how the patient is lying during the MRI examination.
Point 3.p-value test is often used with lower-case “p”
Response 3. This has been corrected, changing all the “p”s to lower case.
Point 4.figures 3 and 5 – impossible to analyse graphs in c) and d) – numbers cannot be seen and lines are very faint, also axes information seems to be missing – please improve these graphics.
Response 4. We have changed the figure numbers as rightly suggested by the reviewer.
Point 5.Following the suggestions above, please provide a systematic study analysing differences in size of ROI between supine and prone exams.
Response 5. The regions of interest are drawn subjectively, obtaining a circle of about 3-5 pixels in diameter inside the tumour to acquire information about the dynamic curve. Given the difference between the prone and supine exams it is not possible to copy the same dimension, so that the ROI dimensions are comparable but not identical. We have taken care to draw them as similar as possible.
Point 6.Line 254: what do you mean by “ROI marked by the number 1”? There is no number 1 marked in the figure
Response 6.The reviewer is absolutely right: the number 1 has been placed adjacent to the circle of each ROI, but since the number is very small and the same color as the ROI it is very difficult to see in both image (a) and image (b).
Point 7.Figure 2 is followed by figure 4, and then figure 3 is followed by figure 5 - why do you not use a regular sequence of numbers?
Response 7.We have changed the numbers of the figures, as correctly suggested by the reviewer.
Point 8.Lines 299-305: a number of observations seem to have been made, but there is no systematic presentation of these in the results. Overall it feels like the data can be a lot more scrutinised and more information should be extracted from the data available to the study.
Response 8. In the results we have presented in sequence all the evaluations made to compare the data obtained in prone and supine positions. We compared the measurements between operators, between different postures and between the different breast lesions. The classification of the lesions was then made using two different methods to consider the dynamic curves. Finally we detected the different representation of the vascularization peaks using the colorimetric maps as presented in Figure 6.
We agree with the reviewer that many other specific observations can certainly be extracted from such a large number of data. In order to make the results section clearer we have added this phrase: “More specifically, Figure 6 shows the greater signal intensity (in red) of the breast lesion, representing a higher contrast impregnation peak compared with the same lesion acquired in supine position.”
Round 2
Reviewer 1 Report
The authors have addressed most of the comments and suggestions, but for the sample size justification please include the calculation and reference as stated in the response:
"It is, however, important to point out that in one of our earlier studies, cited at no. 8 of the bibliography, we observed that the expected prevalence of an additional lesion in breast MRI in prone position appears in 25.3% (490/1930) of total MRIs. Therefore, establishing a level of significance of 5% and an estimate precision of 8%, the minimum sample size is 113 subjects [see Kasiulevičius, V.; Šapoka, V.; Filipavičiūtė, R. Sample size calculation in epidemiological studies. Gerontologija. 2006, 7 (4), 225-231.]. Consequently, the sample of 120 subjects in both prone and supine position available to us proves to be representative of the general population."
In addition, please include response 5 in the manuscript: "...using random numbers generated by computer."
Author Response
Point 1. The authors have addressed most of the comments and suggestions, but for the sample size justification please include the calculation and reference as stated in the response:"It is, however, important to point out that in one of our earlier studies, cited at no. 8 of the bibliography, we observed that the expected prevalence of an additional lesion in breast MRI in prone position appears in 25.3% (490/1930) of total MRIs. Therefore, establishing a level of significance of 5% and an estimate precision of 8%, the minimum sample size is 113 subjects [see Kasiulevičius, V.; Šapoka, V.; Filipavičiūtė, R. Sample size calculation in epidemiological studies. Gerontologija. 2006, 7 (4), 225-231.]. Consequently, the sample of 120 subjects in both prone and supine position available to us proves to be representative of the general population."
Response 1. We agree with the reviewer and we have duly added the phrase in the appropriate section.
Point 2. In addition, please include response 5 in the manuscript: "...using random numbers generated by computer."
Response 2. We agree with the reviewer and we have duly added the phrase in the appropriate section.
Reviewer 2 Report
Very few changes are made, I still believe the Introduction is too short, a number of points could be clarified in the manuscript, the results are very difficult to read and interpret.
Author Response
Point 1. Very few changes are made, I still believe the Introduction is too short, a number of points could be clarified in the manuscript,
Response 1. We have tried to follow the reviewer’s suggestions, increasing the length of the introduction from 298 to 598 words (+84%) in order to present our work as completely as possible.
Point 2. the results are very difficult to read and interpret.
Response 2. The section of our article dealing with the results presents a precise description of all the aspects considered in our study, which consists of a comparison between MRI exams in prone and supine position.
The analysis carried out is wide-ranging and complex. Firstly we considered the parameters and quality of the images, making a comparison. We then set forth the results regarding the number of lesions and the variation in the measurements, demonstrating the presence of differences. Finally we presented the results of the comparison between the classifications, including the vascularisation.
The argument is certainly complex and to this extent we understand the reviewer’s perplexity. Nevertheless, although we appreciate this, we do not share the reviewer’s opinion that the results are difficult to read and interpret. We feel that simplifying the results would not make the presentation clear and effective. The results consist of a series of statistics, tests and tables, which are also commented in the discussion, supporting the conclusions. We trust that the reviewer will comprehend and accept our motives.
Round 3
Reviewer 2 Report
N/A